# Dissociable roles of human frontal eye fields and early visual cortex in presaccadic attention

Nina M. Hanning [1,2] ✉, Antonio Fernández[1,3] & Marisa Carrasco [1]

Shortly before saccadic eye movements, visual sensitivity at the saccade target is enhanced, at the expense of sensitivity elsewhere. Some behavioral and neural correlates of this *presaccadic* shift of attention resemble those of *covert* attention, deployed during fixation. Microstimulation in non-human primates has shown that presaccadic attention modulates perception via feedback from oculomotor to visual areas. This mechanism also seems plausible in humans, as both oculomotor and visual areas are active during saccade planning. We investigated this hypothesis by applying TMS to frontal or visual areas during saccade preparation. By simultaneously measuring perceptual performance, we show their causal and differential roles in contralateral presaccadic attention effects: Whereas rFEF+ stimulation enhanced sensitivity opposite the saccade target throughout saccade preparation, V1/V2 stimulation reduced sensitivity at the saccade target only shortly before saccade onset. These findings are consistent with presaccadic attention modulating perception through cortico-cortical feedback and further dissociate presaccadic and covert attention.

Every time we open our eyes, we confront an overwhelming amount of visual information, yet we have the impression of effortlessly understanding what we see. We are typically not aware of the complex neuro-cognitive processes that help us prioritize visual input. Visual attention, commonly defined as selective processing of certain locations or features, allows us to filter relevant information out of irrelevant noise by focusing on some aspects of the visual scene while ignoring others[1,2]. This attentional selection is frequently achieved by a succession of rapid saccadic eye movements toward relevant information of the visual scene[3]. Interestingly, attention reaches the next location of interest already before the eyes start to move. During saccade preparation, *presaccadic attention* is automatically deployed to the upcoming fixation location, which is indicated by improved visual sensitivity at the saccade target[4–8], at the expense of lowered perceptual sensitivity at other (non-target) locations[9–14]. The perceptual benefits of presaccadic attention rely on feedback from oculomotor structures to visual cortices[15,16] (Fig. 1a). Sub-threshold

microstimulation of the frontal eye fields (FEF) in non-human primates (which would elicit a saccade if stimulated above threshold) modulates activity in visual cortex[17–20] and enhances visual sensitivity at the movement field of the stimulated neurons[21–23] – resembling the typical behavioral correlates of presaccadic attention.

Presaccadic attention modulates the contrast response function (CRF), which characterizes the non-linear, sigmoidal relation between the contrast (or intensity) of a visual stimulus and the resulting response[24–26], such as neuronal firing rate or, consequently, visual performance. Specifically, presaccadic attention alters perceptual performance via response gain, i.e., it increases $d_{max}$ (the maximal response at high contrast) at the saccade target (Fig. 1b), and reduces $d_{max}$ at non-target locations[12,27].

This characteristic 'push-pull mechanism' is also observed for *covert* attentional orienting, in the absence of eye movements: Behaviorally relevant items, to which we voluntarily deploy *endogenous* attention, as well as salient events that automatically capture

[1]Department of Psychology & Center for Neural Sciences, New York University, New York, NY, USA. [2]Institut für Psychologie, Humboldt Universität zu Berlin, Berlin, Germany. [3]Department of Psychology, University of Texas at Austin, Austin, TX, USA. ✉e-mail: hanning.nina@gmail.com

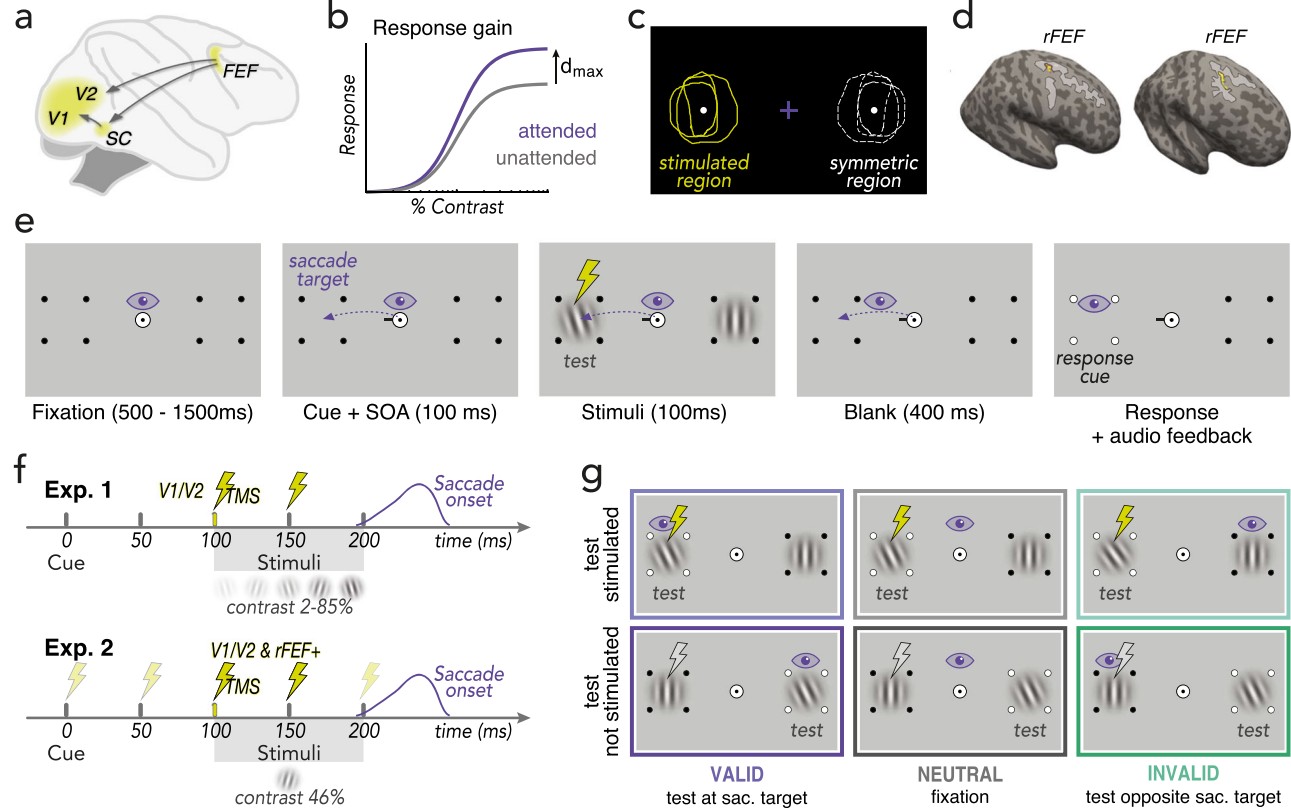

**Fig. 1 | Background and experimental design. a** Feedback connections originally established in non-human primates[15,16,22,115,116] assumed to underlie perceptual correlates of human presaccadic attention. *FEF*: putative human homolog of the Frontal Eye Fields, *SC*: Superior Colliculus; (**b**) Response gain effect of presaccadic attention on the contrast response function: presaccadic attention scales the response by a multiplicative gain factor, resulting in an increase of the asymptotic response $d_{max}$ (the maximal response achieved at high contrast)[12,27]. **c** Determining occipital (V1/V2) stimulation sites (Exp.1 & 2a) and stimulus placement for both experiments via 'phosphene mapping': Observers were stimulated laterally around the occipital pole until they perceived a phosphene (in the contralateral visual field), then drew its outline on the screen. The center of the phosphene drawings (*stimulated region*) and the *symmetric region* (not stimulated) were used for stimulus placement in the main experiment, where we applied sub-phosphene-threshold TMS using identical coil positioning. **d** Determining rFEF+ stimulation sites (Exp.2b). rFEF+ (yellow ROI) was localized on each individual observer's anatomy (two exemplary observers shown here) via a probabilistic topography atlas[82] and verified by anatomical landmarks (junction of the precentral and superior frontal sulcus; light gray areas)[64–67]. **e** Presaccadic orientation discrimination task. After a fixation period, a central direction cue (black line) appeared. Observers were instructed to make a saccade to the indicated target marked by placeholder dots. Note that the saccade was equally likely directed to the

stimulated and to the symmetric region / hemifield. 100 ms after cue onset, a tilted test Gabor patch was presented at either the saccade target (*valid; 50%*) or at the opposite location (*invalid*), randomly intermixed; a vertical Gabor was presented at the other location. Importantly, stimuli were presented during saccade preparation, i.e., while gaze was still at the screen center. After saccade offset, a response cue (white dots on placeholder) indicated the location at which the test Gabor had appeared, and observers reported its orientation. In the *neutral* condition (separately blocked) the cue pointed to both placeholders and observers kept fixating (Supplementary Movie 1 demonstrates the trial sequence of each condition). **f** Trial timeline. Gabor stimuli in both experiments were presented 100–200 ms after cue onset. Observers received double-pulse TMS (50 ms inter-pulse interval) locked to stimuli onset (Exp.1) or at various times during saccade preparation (0–200 ms relative to cue onset; Exp.2). Whereas grating contrast was varied to measure contrast response functions in Exp.1, contrast was fixed to 46% in Exp.2.
**g** Experimental conditions. The test was equally likely presented at the stimulated region (*test stimulated*) or in the opposite hemifield (*not stimulated*). Moreover, the test was equally likely presented at the saccade target (*valid*) or opposite of it (*invalid*). Note that in Exp.2 only the *valid* and *invalid* conditions were tested, for which tilt angles were titrated separately to keep overall task difficulty comparable (*Methods – Titration procedure*).

*exogenous* attention, likewise cause perceptual benefits at the attended and concomitant costs at unattended locations[1,12,26,28,29]. Both covert and presaccadic attention modulate visual processing along the cortical hierarchy. Merely shifting the focus of attention (while keeping the retinal image constant) affects neuronal responses[30–34]. As the perceptual and neuronal dynamics of covert attention seemingly mimic those of saccade preparation, some have postulated that the same neural mechanism underlies the two processes[35–37]; any shift of attention (even during fixation) would result from eye movement planning. Indeed, at a broad scale, oculomotor brain structures such as FEF are also modulated during covert attention[38–40] – yet by distinct sub-populations[30,41–43]. Overall, more recent evidence, contradicts the idea that presaccadic and covert attention would be functionally equivalent and rely on the same neural circuitry[44–49]. Human neuroimaging studies indicate that saccade planning and covert exogenous

and endogenous attention differentially modulate brain activity[50–57]. Yet, these studies cannot establish causality, as fMRI techniques record, but do not manipulate brain function.

Using transcranial magnetic stimulation (TMS) to briefly, and non-invasively, alter cortical activity[58–62], we recently dissociated the causal role of two brain areas involved in covert exogenous and endogenous attention: Whereas occipital stimulation extinguished benefits and costs of exogenous attention[29], it did not alter endogenous attention[63]—which instead was affected by stimulation of rFEF+[63], the putative human homolog of the right macaque frontal eye field[64–67]. Saccade preparation enhances neural responses in oculomotor areas and elicits retinotopic activity in visual cortex[68,69], but it is unknown which brain areas play a *causal* role in presaccadic attention.

The aim of the present study was two-fold: (1) test and potentially dissociate the causal involvement of early visual areas (V1/V2) in

presaccadic attention dynamics from those recently established for covert attention[29,63]; (2) elucidate the differential role of and interplay between frontal (rFEF+) and visual areas (V1/V2) in perceptual benefits and costs preceding saccadic eye movements to investigate the role of cortico-cortical feedback in presaccadic attention. Our findings support the view that presaccadic attention modulates perception through cortico-cortical feedback also in humans, and further dissociate presaccadic and covert attention.

## Results

To assess whether early visual areas are critical for presaccadic attention, in *Experiment 1* we measured CRFs during fixation or saccade preparation in a combined psychophysics-TMS experiment. Saccades were prepared either to the visual field location affected by V1/V2 TMS, individually determined using an established phosphene mapping procedure[29,63,70,71] (Fig. 1c; Methods – Phosphene mapping & stimulus placement), or to the symmetric region in the other hemifield (internal control, unaffected by TMS). Shortly before saccade onset, we briefly presented oriented test stimuli either at or opposite to the saccade target (Fig. 1e) and applied (sub-threshold) double-pulse TMS to the individually determined V1/V2 stimulation site (Fig. 1f). We tested multiple stimulus contrasts and derived CRFs for all combinations of presaccadic attention at the stimulated and non-stimulated symmetric location (Fig. 1g), to characterize the effect of TMS on presaccadic performance benefits and costs relative to fixation.

### Differential role of early visual cortex in presaccadic and covert attention

To assess whether TMS to early visual areas affects presaccadic attentional modulations, we obtained CRFs by fitting Naka-Rushton functions[24] to the visual sensitivity data (Fig. 2a; Methods – Quantification & statistical analysis). To evaluate the predicted response gain effect of presaccadic attention on the upper asymptote $d_{max}$[12,27] as well as the semi-saturation contrast $C_{50}$ and a potential effect of V1/V2 TMS on either, we conducted two repeated measures ANOVAs [presaccadic attention (valid/neutral/invalid) * TMS side (test stimulated/not stimulated)]. $C_{50}$ was modulated by presaccadic attention ($F(2,18) = 8.95$, $p = 0.002$), but not by TMS side ($F(1,9) = 4.03$, $p = 0.076$; BF[0.50:1],

$p$BIC($H_0$|D) = 0.33, $p$BIC($H_1$|D) = 0.67) or the interaction between attention and TMS side ($F(2,18) = 2.15$, $p = 0.145$; BF[2.34:1], $p$BIC($H_0$|D) = 0.70, $p$BIC($H_1$|D) = 0.30). Presaccadic attention also modulated the upper asymptote of the functions ($F(2,18) = 164.34$, $p < 0.001$), which, compared to *neutral* (fixation baseline), was significantly increased at the saccade target (*valid*; $p < 0.001$, *Cohen's d* = 1.82) and decreased at the non-target (*invalid*; $p < 0.001$, *Cohen's d* = 5.26) – reflecting the typical presaccadic benefit and cost, respectively. Importantly, TMS side neither affected asymptotic performance ($F < 1$; BF[1.98:1], $p$BIC($H_0$|D) = 0.66, $p$BIC($H_1$|D) = 0.34), nor did it interact with presaccadic attention ($F < 1$; BF[7.80:1], $p$BIC($H_0$|D) = 0.89, $p$BIC($H_1$|D) = 0.11).

This finding is in direct contrast to the effect of covert exogenous attention[29]: In the same psychophysics-TMS protocol (except that instead of a central saccade cue, a peripheral attention cue modulated attention), V1/V2 TMS eliminated response gain benefits and costs of covert attention (Fig. 2b, exogenous attention and Supplementary Fig. 1a). Covert endogenous attention (modulated by a central cue indicating the likely test location), however, like presaccadic attention, was not affected by V1/V2 TMS[63] (Fig. 2b, endogenous attention and Supplementary Fig. 1b). These results reveal a neural dissociation of presaccadic attention and covert exogenous attention (measured previously[29]), as verified by a significant 3-way interaction among attention type (presaccadic/exogenous attention; between-subject factor), attention condition (valid/neutral/invalid), and TMS side (test stimulated/not stimulated), $F(2,36) = 4.63$, $p = 0.016$. Moreover, they show that the perceptual effects of presaccadic attention (i.e., the difference between *valid* and *invalid*) are stronger than those of previously measured covert exogenous[29] and endogenous[63] attention (Fig. 2b). This also applies to the separate benefits (difference *valid* and *neutral*) and costs (difference *invalid* and *neutral*), which are both more pronounced for presaccadic attention (Fig. 2a) than for covert exogenous[29] (Supplementary Fig. 1a) and endogenous[63] (Supplementary Fig. 1b) attention.

When investigating the role of early visual areas in covert attention[29,63], we applied V1/V2 TMS at the known peak of exogenous and endogenous attention effects on performance[1,32,72], i.e., 100 ms[29] and 500 ms[63] after the respective cue onset. Presaccadic attention

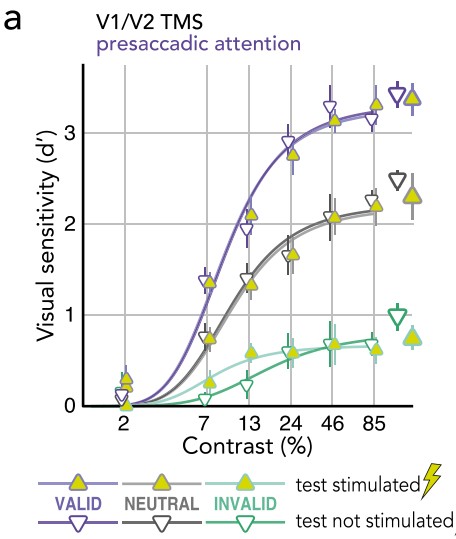

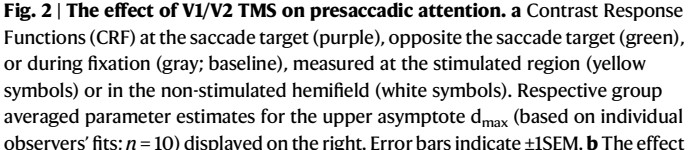

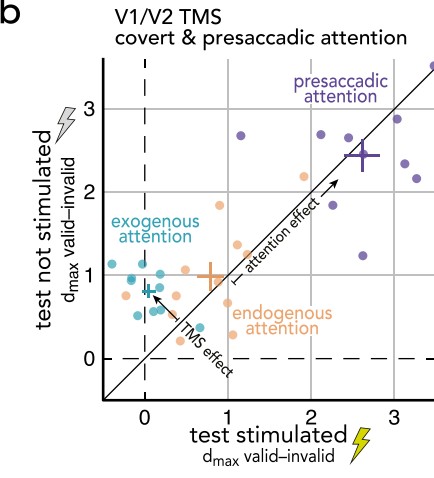

**Fig. 2 | The effect of V1/V2 TMS on presaccadic attention. a** Contrast Response Functions (CRF) at the saccade target (purple), opposite the saccade target (green), or during fixation (gray; baseline), measured at the stimulated region (yellow symbols) or in the non-stimulated hemifield (white symbols). Respective group averaged parameter estimates for the upper asymptote $d_{max}$ (based on individual observers' fits; $n = 10$) displayed on the right. Error bars indicate ±1SEM. **b** The effect of V1/V2 TMS on presaccadic (purple; $n = 10$) and covert exogenous (blue; $n = 10$, data previously published[29]) and endogenous attention (orange; $n = 12$, data previously published[63]). Dots represent individual observers' attention effects (valid minus invalid $d_{max}$ estimates) at the stimulated (x-axis) plotted against the not stimulated region (y-axis). Crosses represent the group mean ±1SEM. Source data are provided as a Source Data file.

builds up during saccade preparation, gradually reaching its maximum shortly before saccade onset—which typically is ~200 ms after saccade cue onset[7–9,73–76], but this varies with the difference in saccade latencies among and within individual observers. To investigate the role of early visual areas throughout saccade preparation (including the time of peak presaccadic attention right before saccade onset), in *Experiment 2a* we stimulated V1/V2 at various timepoints after saccade cue onset (Fig. 1f), using the otherwise identical psychophysics-TMS protocol.

Neurophysiological evidence in non-human primates shows that saccade preparation modulates neural and perceptual sensitivity via cortico-cortical feedback from higher-order to early visual areas[15–23]. EEG and concurrent fMRI-TMS studies indicate that similar feedback projections seem to exist in humans: Activity in FEF+ precedes occipital activation during saccade preparation[77,78] and FEF + TMS modulates activity in visual cortex[79–81] and enhances perceived contrast in the contralateral hemifield[79]. Following the assumption of presaccadic cortico-cortical feedback, V1/V2 should contribute to (and thus V1/V2 TMS should affect) presaccadic attention at later stages of saccade programming, whereas TMS of the FEF—an area providing the source of feedback, which we stimulated in *Experiment 2b* – should show a relatively earlier effect. V1/V2 stimulation site and stimulus placement were determined via phosphene mapping[29,63,70,71] (Fig. 1c). We localized rFEF+, the right putative human homolog of macaque frontal eye field[64–67], on each observer's anatomical brain scan (acquired via MRI) via a probabilistic topography atlas[82] (Fig. 1d; Methods – FEF Localization). Previous TMS studies investigating the role of FEF+ on attentional modulations have found that predominantly the right FEF+ affects behavior[83–85]. We assume a hemifield-specific, contralateral effect of rFEF+ stimulation[86], as we have previously shown that with the same neuro-navigated rFEF+ localization and stimulation protocol (inter-pulse timing and intensity), TMS only affected perceptual modulations of covert endogenous attention in the contralateral hemifield[63]—leaving the known effects on visual performance in the ipsilateral hemifield unaffected (but see[80] for contra- and ipsilateral modulation of sensitivity in area MT/V5 by rFEF stimulation).

For a temporal analysis of the effect of V1/V2 and rFEF+ TMS on saccade latencies and landing precision, see Supplementary Fig. 2. Overall, presaccadic TMS slowed down saccadic reaction times: The later the double-pulse was applied (after the saccade cue), the longer the saccade latencies. This effect, however, was not specific to the stimulation site and likewise occurred for saccades to the stimulated and unstimulated hemifields, suggesting a general alerting rather than stimulation specific effect. Landing precision was not affected by V1/V2 or rFEF+ TMS at any tested timepoint.

## Visual areas V1/V2 causally modulate presaccadic benefits shortly before saccade onset

To investigate the causal role of early visual areas on the effects of presaccadic attention throughout saccade preparation, we evaluated visual sensitivity at the saccade target –where presaccadic attention *benefits* performance– and opposite of the saccade target –where presaccadic attention impairs performance– as a function of V1/V2 stimulation time relative to saccade onset (Methods – Quantification & statistical analysis). Given that the saccade target either matched the V1/V2 stimulated region (Fig. 1c; Methods – Phosphene mapping & stimulus placement) or the 'symmetric region' unaffected by TMS (internal control), by comparing these two sides we can directly assess the effect of V1/V2 TMS on the perceptual benefits (at the saccade target) and costs (opposite the saccade target) caused by presaccadic attention.

We conducted a repeated measures ANOVA with the factors presaccadic attention (valid/invalid), TMS side (test stimulated/not stimulated), and TMS time ($175 \pm 25$ ms/$125 \pm 25$ ms/$75 \pm 25$ ms/$25 \pm 25$ ms prior to saccade onset). Visual sensitivity was higher in valid than invalid trials ($F_{(1,8)} = 6.44$, $p = 0.035$). Main effects of TMS side

($F < 1$; BF[2.41:1], $p$BIC(H$_0$|D) = 0.71, $p$BIC(H$_1$|D) = 0.29) and TMS time ($F < 1$; BF[48.35:1], $p$BIC(H$_0$|D) = 0.98, $p$BIC(H$_1$|D) = 0.02) were not significant, and neither were any of the two-way interactions (all $p > 0.094$; BF[2.67–89.13:1], $p$BIC(H$_0$|D) = 0.73–0.99, $p$BIC(H$_1$|D) = 0.01–0.27). However, a significant 3-way interaction ($F_{(3,24)} = 3.89$, $p = 0.026$) emerged because V1/V2 TMS significantly reduced presaccadic benefits at the saccade target (compared to when the saccade target was not stimulated) when applied within the last 50 ms prior to saccade onset ($p = 0.008$, *Cohen's d* = 1.08 Fig. 3a–left). Thus, early visual areas become crucial for presaccadic benefits in the final stage of saccade programming, shortly before saccade onset – when presaccadic attention reaches its maximum effect[7–9,73–76]. Presaccadic costs opposite the saccade target (Fig. 3a–right) were not affected by V1/V2 TMS at any timepoint. A repeated measures ANOVA [presaccadic attention (valid/invalid) * TMS time] showed that within 50 ms before saccade onset, the effect of V1/V2 TMS (computed as test stimulated–not stimulated; Fig. 3b) on sensitivity at the saccade target (valid) was significantly stronger than opposite of it (invalid) (attention * TMS time: $F_{(3,24)} = 3.89$, $p = 0.021$; valid vs. invalid at $25 \pm 25$ ms: $p = 0.047$, *Cohen's d* = 0.51) – documenting that early visual areas play a selective causal role for presaccadic benefits only shortly before saccade onset.

## rFEF+ stimulation reduces presaccadic costs throughout saccade preparation

To assess the contribution of human FEF to the effects of presaccadic attention throughout saccade preparation, we stimulated rFEF+ and observed a different pattern from that of V1/V2 stimulation, using the otherwise identical experimental protocol, analysis and participant sample (though two observers were no longer available). The repeated measures ANOVA [presaccadic attention (benefits/costs) * TMS side (test stimulated/not stimulated) * TMS time ($175 \pm 25$ ms to $25 \pm 25$ ms before saccade onset) showed no significant main effect of attention ($F_{(1,6)} = 1.11$, $p = 0.332$; BF[1.40:1], $p$BIC(H$_0$|D) = 0.58, $p$BIC(H$_1$|D) = 0.42) or stimulation time ($F_{(3,18)} = 2.65$, $p = 0.110$; BF[1.01:1], $p$BIC(H$_0$|D) = 0.50, $p$BIC(H$_1$|D) = 0.50), but a significant main effect of TMS side ($F_{(1,6)} = 12.90$, $p = 0.011$). TMS side also interacted with presaccadic attention ($F_{(1,6)} = 8.23$, $p = 0.028$), which is explained by increased visual sensitivity at the (stimulated) location opposite the saccade target caused by rFEF+ stimulation throughout saccade preparation, i.e., rFEF+ TMS reduced presaccadic costs; $F_{(1,6)} = 14.72$, $p = 0.009$, *Cohen's d* = 1.45; Fig. 3c–right). In contrast, rFEF+ TMS did not affect sensitivity at the (stimulated) saccade target ($F < 1$, $p = 0.778$; BF[2.52:1], $p$BIC(H$_0$|D) = 0.72, $p$BIC(H$_1$|D) = 0.28; Fig. 3c–left). Neither the 3-way interaction was significant ($F_{(3,18)} = 1.44$, $p = 0.269$; BF[7.71:1], $p$BIC(H$_0$|D) = 0.89, $p$BIC(H$_1$|D) = 0.11), nor did presaccadic attention ($F_{(3,18)} = 1.76$, $p = 0.200$; BF[4.33:1], $p$BIC(H$_0$|D) = 0.81, $p$BIC(H$_1$|D) = 0.19) or TMS side ($F < 1$; BF[75.72:1], $p$BIC(H$_0$|D) = 0.99, $p$BIC(H$_1$|D) = 0.01) interact with stimulation time. A repeated measures ANOVA [presaccadic attention (valid/invalid) * TMS time] on the effect of rFEF+ TMS (again computed as test stimulated–not stimulated; Fig. 3d) showed a main effect of presaccadic attention ($F_{(1,6)} = 8.23$, $p = 0.028$); the main effect of stimulation time ($F < 1$; BF[59.57:1], $p$BIC(H$_0$|D) = 0.98, $p$BIC(H$_1$|D) = 0.02) and its interaction with presaccadic attention ($F_{(3,18)} = 1.44$; $p = 0.268$; BF[10.07:1], $p$BIC(H$_0$|D) = 0.91, $p$BIC(H$_1$|D) = 0.09) were not significant. The effect of rFEF+ stimulation on presaccadic costs (enhancing sensitivity opposite the saccade target) thus was stronger than on presaccadic benefits at the saccade target, independent of stimulation time.

We note that the overall sensitivity difference between the invalid not-stimulated conditions of Exp.2a (Fig. 3a) and Exp.2b (Fig. 3c) –for which we expect no influence of TMS stimulation– can be explained by our tilt angle adjustment procedure: If overall performance in valid or invalid trials (across hemifields) deviated from 80%, we slightly adjusted the tilt angle of the respective condition accordingly

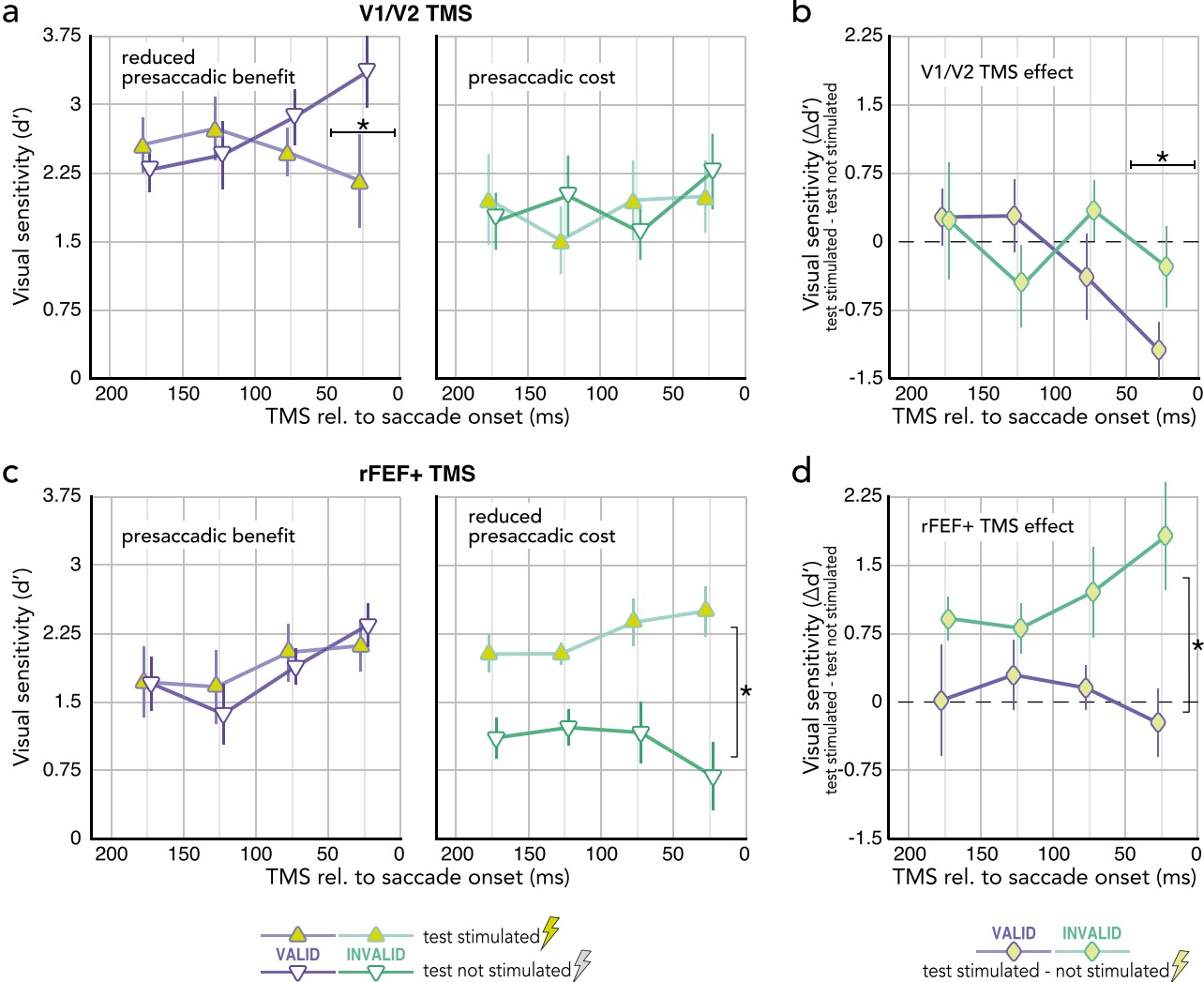

**Fig. 3 | The effect of occipital TMS (V1/V2, Exp.2a, *n* = 9; upper row) and frontal TMS (rFEF+, Exp.2b, *n* = 7; lower row) on presaccadic benefits and costs throughout saccade preparation. a** Visual sensitivity at the saccade target (purple) and opposite the saccade target (green) measured at the region matching V1/V2 stimulation (yellow triangles) or opposite of it (white triangles, control), binned as a function of stimulation time relative to saccade onset. **b** The effect of V1/V2 TMS on presaccadic benefits at the saccade target (purple; valid test stimulated−not stimulated) and *costs* opposite the saccade target (green; *invalid* test stimulated−not stimulated) binned as a function of stimulation time relative to saccade onset. **c** The effect of rFEF+ TMS on visual sensitivity at the saccade target and opposite of it across time; conventions as in (**a**). **d** The effect of rFEF+ TMS on presaccadic benefits and costs across time; conventions as in (**b**). All symbols and error bars represent the group average ±1SEM. Asterisks indicate significant differences between the two compared conditions at a respective time point (**a,b**; Bonferroni corrected two-sided post-hoc comparison after significant ANOVA 3-way interaction) or between the two compared conditions across time (**c,d**; significant ANOVA main effect). *p < 0.05. Source data are provided as a Source Data file.

(see Methods – Titration procedure). Given that rFEF+ stimulation increased performance in the stimulated hemifield (and thus overall performance), the resulting tilt angle adjustment to keep overall task difficulty in the invalid condition around 80% may have lowered performance in the unstimulated hemifield.

There was a diverging (although non-significant) trend in sensitivity time course across TMS sites in the invalid not-stimulated conditions (Fig. 3a, c), which might reflect a push-pull mechanism: V1/V2 stimulation reduced sensitivity at the saccade target in the stimulated hemifield right before saccade onset, leaving more (attentional) resources for other locations, such as the (invalid) location opposite the saccade target (in the non-stimulated hemifield). rFEF+ stimulation did not affect sensitivity at the saccade target, causing no such "enhancement" at the invalid location (in the non-stimulated hemifield).

Combined, the results of Experiment 2 demonstrate a differential involvement of occipital and frontal areas in presaccadic attention.

Whereas rFEF+ stimulation reduced presaccadic costs (i.e., improved the typically low sensitivity) at non-saccade targets throughout saccade preparation, V1/V2 stimulation reduced presaccadic benefits (i.e., reduced the typically high sensitivity) at the saccade target when applied just before saccade onset. This dissociation is reflected in a 2 (stimulation site; V1/V2 vs. rFEF +) * 2 (attention condition; valid vs. invalid) * 4 (stimulation time) repeated measures ANOVA, in which we compared presaccadic benefits and costs at the V1/V2 or rFEF + stimulated hemifield for the six observers that participated in both Experiment 2a and 2b. The stimulation site (V1/V2 vs. rFEF +) interacted with attention condition (valid / benefits vs. invalids / costs), $F_{(1,5)} = 6.72$, $p = 0.048$. This was not the case at the ipsilateral, not stimulated hemifield (stimulation site * attention conditions: $F < 1$; BF[2.35:1], $p$BIC(H$_0$|D) = 0.70, $p$BIC(H$_1$|D) = 0.30), further indicating that the respective effects of V1/V2 and rFEF+ stimulation on presaccadic benefits and costs were specific to the stimulated hemifield.

## Discussion

This is the first saccade study to investigate the effects of V1/V2 and FEF+ TMS using the same experimental design and stimulation protocol, which enables us to compare the causal role of these brain regions in presaccadic attention. In two psychophysics-TMS experiments we dissociated the involvement of early visual cortex (V1/V2) in presaccadic and covert exogenous[29] and endogenous attention[63] (deployed in the absence of eye movements), and document the differential role of frontal (rFEF+) and visual areas (V1/V2) as well as their interplay in presaccadic attention.

V1/V2 TMS, at a fixed time (-150 ms) after cue onset, did not affect presaccadic attention – unlike covert exogenous attention, which is extinguished[29] with the same psychophysics protocol and TMS pulse timing (Fig. 2 and Supplementary Fig. 1). V1/V2 TMS only affected presaccadic attention just before saccade onset (Fig. 3a). This pattern of results shows that both exogenous and presaccadic attention recruit early visual areas, but the time at which these areas play a critical role differs. Our results thus causally dissociate the two forms of attentional orienting, providing further evidence against the claim that covert *exogenous* attention is functionally equivalent to oculomotor programming[87–89]. The observed effects of presaccadic attention seem similar to those of covert *endogenous* attention[63], in that they both are affected by FEF+ stimulation. However, perceptual benefits and costs of presaccadic attention are stronger than those of covert endogenous (and exogenous) attention (Fig. 2b). Moreover, rFEF+ TMS both eliminated the benefits and reduced the costs of endogenous attention[63], but it only reduced presaccadic costs at the non-target location, leaving (the typically high) presaccadic sensitivity at the saccade target unaffected (Fig. 3c).

The rFEF+ TMS induced enhancement opposite the saccade target is consistent with evidence that TMS effects depend on the brain activation state at the time of stimulation[29,63,90–96]: TMS is more likely to increase performance where it is usually worse (e.g., opposite the saccade target). It is also in line with the view of FEF being a priority map, controlling / shifting the focus of attention[15,16,97]. Similarly, FEF-microsimulation in non-human primates increases visual sensitivity at the movement field of the stimulated neurons[21–23], akin to a shift of attention.

Previous TMS-studies investigating the influence of FEF+ on presaccadic attention have yielded mixed results, reporting that FEF+ TMS either decreased[98] or increased[99] sensitivity at the saccade target. Crucially, these studies have not evaluated the TMS-effect relative to saccade onset (as we do in Experiment 2). However, the effects of presaccadic attention gradually increase throughout saccade preparation[49] and saccadic reaction times vary pronouncedly between (and within) individual observers – which is why presaccadic attention effects typically are evaluated relative to saccade onset[7,74–76]. The inconsistent results of previous TMS studies[98,99] could be explained by FEF+ stimulation being applied at different time points relative to saccade onset (i.e., when the effects of presaccadic attention would be differently pronounced).

V1/V2 TMS shortly (within 50 ms) before saccade onset, i.e., at the peak of presaccadic attention, reduced the typical sensitivity benefit at the saccade target (Fig. 3a). Importantly, this effect was time-locked to saccade onset, indicating that occipital regions are recruited right before the eyes move. Applied at a fixed time relatively earlier during saccade programming (100 ms after cue onset), V1/V2 TMS did not affect presaccadic attention (Fig. 2a).

To conclude, our results dissociate the neural basis of presaccadic from covert attention and demonstrate a causal and differential role of occipital and frontal areas in presaccadic benefits (at the saccade target) and costs (opposite of it). Whereas the effect of frontal TMS was present throughout saccade preparation, critically, the effect of occipital TMS was locked to the period right before saccade onset, which – consistent with presaccadic feedback from oculomotor structures to visual cortex[15,16,22] – reveals that occipital regions are recruited only during later stages of saccade programming.

## Methods

### Observers

Ten observers (7 female; aged 22–36 years) participated in Experiment 1. Ten observers (8 of which participated in Exp.1; 7 female; aged 21–36 years) participated in Experiment 2a, of which 7 observers (4 female; aged 22–36 years) also participated in Experiment 2b. Data of one observer in Experiment 2a were not included in the analysis due to an insufficient number of trials in the earliest time bin. Note that the pattern of results and reported statistical (null-)effects were the same with or without this exclusion. All observers had normal or corrected-to-normal vision, provided written informed consent, and (except for one author) were naive to the purpose of the experiment. We chose a sample size in the range of previous psychophysics-TMS studies investigating presaccadic and covert attention[29,63,98,99]. Observers were screened for TMS contraindications prior to participation. The protocols for the study were in accordance with the safety guidelines for TMS research and approved by the University Committee on Activities Involving Human Subjects at New York University and all experimental procedures were in agreement with the Declaration of Helsinki.

### Setup

Observers sat in a dark room with their head stabilized by a chin and forehead rest and viewed the stimuli at 57 cm distance on a gamma-linearized ViewPixx/EEG LCD monitor (VPixx Technologies, Saint-Bruno, QC, Canada) with a spatial resolution of 1920 by 1080 pixels and a vertical refresh rate of 120 Hz. Gaze position of the dominant eye was recorded using an EyeLink 1000 Desktop Mount eye tracker (SR Research, Osgoode, ON, Canada) at a sampling rate of 1 kHz. Manual responses were recorded via a standard keyboard. A Linux desktop machine running Matlab (MathWorks, Natick, MA, USA) with Psychophysics[100,101] and EyeLink toolboxes[102] controlled stimulus presentation and response collection.

### Transcranial magnetic stimulation (TMS) and neuronavigation

Observers were stimulated using a 70-mm figure-of-eight coil positioned over the occipital (Exp.1 & Exp.2a) or frontal (Exp.2b) cortex with the handle oriented perpendicular to the sagittal plane. TMS pulses were applied using a 3.5 T Magstim Rapid Plus stimulator (Plymouth, MN, USA) and triggered with MATLAB using an Arduino board Uno (Turin, Italy). In Experiment 1 and Experiment 2a stimulation threshold was defined as the machine intensity required for an observer to perceive a phosphene 50% of the time (mean intensity: 61.2 ± 3.1% (Exp.1) and 62.4 ± 1.8% (Exp.2a) of the maximum stimulator output). In Experiment 2b, stimulation intensity was fixed at 65% of maximum stimulator output. Stimulation intensity remained constant across experimental sessions.

### Phosphene mapping and stimulus placement

Prior to Experiment 1, observers fixated a fixation target at the center of the black screen. We applied a train of seven TMS pulses (30 Hz, 65% of maximal stimulator output) at the assumed phosphene region (laterally around the occipital pole). Once observers perceived a reliable phosphene in the contralateral visual field, they drew its outline on the screen using a computer mouse, and the exact TMS coil position and angle was recorded using the Brainsight TMS navigation system (Rogue Research, Montréal, QC, Canada). The center of each observer's phosphene drawing was used for stimulus placement in Experiments 1 and 2 (see Fig. 1c) – where we stimulated the same region, but with verified sub-threshold stimulation intensity (i.e., observers did not perceive phosphenes during the main experiments), to not contaminate the measure and capture attention 'visually'[103]. TMS coil positions eliciting phosphenes were validated before each

experimental session. In Experiment 1, 7 observers perceived phosphenes in the right visual field and three observers perceived phosphenes in the left visual field (average eccentricity from fixation: 6.39° ± 0.63°; mean±1SEM). In Experiment 2a, 6 observers perceived phosphenes in the right visual field and four observers perceived phosphenes in the left visual field (average eccentricity 6.68° ± 0.66°). For the analysis of behavioral results, we collapse data across phosphene sides, as previous studies stimulating V1/V2 have found no side-specific effects[29,63,71].

## FEF localization
Before participating in Experiment 2a, we localized each observer's (human) right Frontal Eye Field (rFEF + ) using the Wang atlas[82], which has been shown to be a reliable indicator of FEF+ [63,104,105]. We mapped the right FEF+ onto each observer's native volume using mri_surf2vol & mri_surf2surf in Freesurfer[106] (Fig. 1d). The rFEF+ region of interest (ROI) was validated via anatomical landmarks – the junction of the precentral and superior frontal sulci[64,65,67]. Observers' individual anatomical brain scans (T1 image) and rFEF+ ROI were loaded into the neuro-navigation software Brainsight (Rogue Research, Montréal, QC, Canada) for precise stimulation of rFEF+ in Experiment 2b.

## Experimental design
*Experiment 1* (see Fig. 1e and Supplementary Video 1). Observers fixated a central fixation target comprising a black (~0 cd/m2) and white (~96 cd/m2) bull's-eye (r = 0.3°) on gray background (~48 cd/m2). Two placeholders indicated the two potential saccade target locations left and right of fixation, each comprised four black dots (r = 0.15°). Saccade target centers were determined for each observer via phosphene mapping (see *Phosphene mapping & stimulus placement*). Once stable fixation was detected within a 1.75° from fixation for at least 250 ms, the trial started with a jittered fixation period (250 ms, 750 ms, or 1250 ms), before a central direction cue (black line, length 0.75°) pointed to one of the placeholders, thereby cueing the saccade target. Observers were instructed to "look as fast and precisely as possible" to the center of the indicated placeholder. Note that the saccade was equally likely directed to the *stimulated* phosphene region and to the non-stimulated *symmetric* region (Fig. 1c). 100 ms after saccade cue onset (i.e., within the movement latency, gaze still rests at fixation), Gabor gratings (2cpd, random phase) appeared for 100 ms within each placeholder; one grating was vertical, the other *test* grating was slightly tilted (see *Titration procedure*). Gabor contrast varied from trial to trial (method of constant stimuli). Gabor size was adjusted according to the Cortical Magnification Factor: $[M = M0(1 + 0.42E + 0.000055E3)-1]$[107], where M0 refers to the cortical magnification factor (7.99 mm/deg) and E to the stimulus eccentricity in degrees of visual angle; Gabors were scaled to match a cortical magnification of a 2° wide Gabor at 4° eccentricity. Placeholder dots were separated by 1° from the Gabors.

The first TMS pulse was time-locked to Gabor onset, followed by another pulse 50 ms later. 400 ms after Gabor offset (the eye movement has now been performed), the dots of one placeholder changed color from black to white, functioning as a response cue to indicate the location that had contained the tilted Gabor. Observers indicated their orientation judgment via button press (clockwise or counterclockwise, two-alternative forced choice) and were informed that the orientation report was non-speeded. They received auditory feedback for incorrect responses. Importantly, the tilted test Gabor was equally likely presented at the saccade target (*valid* trials) or at the opposite, non-target location (*invalid* trials), i.e., the saccade cue was not predictive of the test location; valid and invalid trials were randomly intermixed. In separately blocked *neutral* trials, two central line cues indicated both placeholder locations and participants were instructed to keep fixating. After an initial training without TMS, observers performed 18 experimental blocks (12 saccade blocks with valid and invalid trials, 6 fixation blocks with neutral trials); random order, 160 trials per block)

split into 3 experimental sessions. We controlled online for broken eye fixation (outside 1.75° from central fixation before the cue onset), too short (<150 ms) or too long (>500 ms) eye movement latencies, and imprecise saccades (not landing within 2.5° from saccade target center). Erroneous trials were repeated in random order at the end of each block.

In *Experiment 2* we used the same psychophysics-TMS as in Experiment 1, with the following differences: (1) Double-pulse V1/V2 (Exp.2a) or rFEF+ (Exp.2b; see *FEF Localization*) TMS was applied at different time points throughout saccade preparation (first TMS pulse 0–200 ms relative to cue onset; Fig. 1f); Gabor timing, as in Experiment 1, was fixed (100–200 ms after cue onset). (2) Whereas grating contrast in Experiment 1 was varied to measure CRF, the contrast was fixed to 46% in Experiment 2. (3) We only tested *valid* and *invalid* trials (randomly intermixed), with separately determined Gabor tilt angles (see *Titration procedure*). Note that stimulus placement was individually determined for each observer (see *Phosphene mapping & stimulus placement*), but identical for both Experiment 2a and 2b. Observers performed 7 (Exp.2a) or 10 (Exp.2b) experimental blocks of 160 trials each, split into 2 experimental sessions. Note that we increased the number of blocks in Experiment 2b to be able to account for a potential loss of trials due to an effect of rFEF+ stimulation on saccade parameters. We again repeated trials with broken eye fixation (outside 1.75° from central fixation before the cue onset), too short (<150 ms) or too long (>500 ms) eye movement latencies, or imprecise saccades landing further than 2.5° from saccade target center.

In total, we included 22,679 trials in the analysis of the behavioral results of Experiment 1 (on average 2268 ± 35 (mean±1SEM) trials per observer), 5295 (530 ± 13) trials in the analysis of Experiment 2a, and 5868 (838 ± 31) trials in the analysis of Experiment 2b.

## Titration procedure
To match overall task difficulty to each observer's visual sensitivity and to account for any learning effects, we titrated the Gabor-tilt angle (±0.5°–6° relative to vertical) separately for each observer before each experimental session (without TMS) via an adaptive staircase procedure[108] implemented in the Palamedes toolbox[109]. For Experiment 1, we used the procedure of the *neutral* condition to determine the tilt angle at which observers' orientation discrimination performance at the highest contrast level (85%) was -$d'$ = 2. This tilt angle (group average 1.7° ± 0.2°) was used in the main experiment for the *valid*, *invalid*, and *neutral* conditions. For Experiment 2, to account for the observed pronounced sensitivity differences at and opposite the saccade target, we titrated the tilt angle during saccade preparation separately for *valid* (Exp.2a: 1.1° ± 0.1°; Exp.2b: 1.1° ± 0.2°) and *invalid* (Exp.2a: 4.1° ± 1.0°; Exp.2b: 4.0° ± 1.5°) trials using a Gabor contrast of 46% (same contrast as in main experiment). In Experiment 2, if the overall discrimination performance in valid or invalid trials (across hemifields) deviated more than ±10% from 80% (-$d'$ = 2), we slightly adjusted the tilt angle of the respective condition accordingly.

## Eye data preprocessing
We scanned the recorded eye-position data offline and detected saccades based on their velocity distribution[110] using a moving average over 20 subsequent eye position samples. Saccade onset and offset were detected when the velocity exceeded or fell below the median of the moving average by 3 standard deviations for at least 20 ms. We included trials in which no blink occurred during the trial and correct eye fixation was maintained within a 1.75° radius centered on central fixation throughout the trial (fixation trials) or until cue onset (saccade trials). Moreover, we only included those eye movement trials in which the initial saccade landed within 2.0° from the required target location and in which the test signal was presented within 100 ms before saccade onset (i.e., the saccade started only after test signal presentation, but not later than 100 ms after signal offset).

## Quantification and statistical analysis

Task performance, indexed by visual sensitivity [d-prime; $d' = z$(hit rate) - $z$(false alarm rate)], was measured as a function of stimulus contrast using the method of constant stimuli (6 Michelson contrast levels: 2, 7, 13, 24, 46, 85%). We arbitrarily defined counter-clockwise responses to counter-clockwise oriented gratings as hits and counter-clockwise responses to clockwise oriented gratings as false-alarms[12,27,29,63,111]. To avoid infinite values when computing $d'$, we substituted hit and false alarm rates of 0 and 1 by 0.01 and 0.99, respectively[8,45,46,112].

To obtain CRF for each condition and test location we fit each observer's data with Naka-Rushton functions[24], parameterized as $d'(C) = d_{max} C^n / (C^n + C_{50}^n)$, where C is the contrast level, $d_{max}$ is the asymptotic performance, $C_{50}$ is the semi-saturation constant (contrast level corresponding to half the asymptotic performance), and $n$ determines the slope of the function. The error was minimized using a least-squared criterion; $d_{max}$ and $C_{50}$ were free parameters, $n$ was fixed. Contrast levels were log-transformed prior to fitting. A change in $d_{max}$ indicates a response gain change, a change in $C_{50}$ a contrast gain. Overall goodness of fit: $R^2 = 0.76 \pm 0.04$.

We used repeated measures ANOVAs to assess statistical significance, followed by Bonferroni-corrected multiple (post-hoc) comparisons, if applicable. For repeated-measures ANOVAs in which the sphericity assumption was not met, we report Greenhouse-Geisser corrected $p$-values. To follow-up statistical null effects, we estimated Bayes factors (BF) as well as Bayesian information criterion probabilities (pBICs) for the null $H_0$ and alternative $H_1$ hypotheses based on the respective ANOVA partial eta squared ($\eta_p^2$)[113]. A Bayes factor greater than 3 provides additional support for the null hypothesis[114].

For Experiment 2, we binned trials as a function of the time between Gabor stimuli offset and saccade onset (see *Eye data pre-processing*) in four separate time windows (200–150 ms, 150–100 ms, 100–50 ms, and 50–0 ms).

## Reporting summary

Further information on research design is available in the Nature Portfolio Reporting Summary linked to this article.

## Data availability

Raw eye tracking and behavioral data is available from the OSF database at https://osf.io/pcunw/. Source data are provided with this paper.

## Code availability

Analysis code to generate manuscript figures is available from the OSF database at https://osf.io/pcunw/.

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

## Acknowledgements

This research was supported by a Marie Skłodowska-Curie individual fellowship (MSCA-IF 898520) by the European Commission to N.M.H., an NIH NINDS grant (F99-NS-120705) to A.F., and an NIH NEI grant (RO1-EY-019693) to M.C. We thank the Carrasco Lab members, in

particular Yuna Kwak and Hsing-Hao Lee, as well Ilona Bloem and Jan Kurzawski for helpful comments and discussions.

## Author contributions
Conceptualization: N.M.H. and M.C.; Methodology and software: N.M.H.; Investigation: N.M.H.; Formal analysis: N.M.H. and A.F.; Visualization: N.M.H.; Writing – original draft: N.M.H.; Writing – review & editing: N.M.H., A.F. and M.C.; Funding acquisition: N.M.H. and M.C.

## Competing interests
The authors declare no competing interests.
