## [Peer Review File · Nature Communications]

Dissociable roles of human frontal eye fields and early visual cortex in presaccadic attentionREVIEWER COMMENTS

Reviewer #1 (Remarks to the Author):

This study investigates the feedback mechanism between oculomotor structures and visual cortices in humans during the presaccadic period, during which visual sensitivity is enhanced at the saccade target and reduced at non-target locations. Using transcranial magnetic stimulation (TMS) on frontal (rFEF+) or visual (V1/V2) areas during saccade preparation and measuring perceptual performance, the authors found differential effects of rFEF+ and V1/V2 stimulation on presaccadic benefits and costs. By comparing the results with existing datasets, they further claim a dissociation of areas involved in presaccadic and covert attention.

The study is important, very well written and the results are presented clearly. The analyses in general are conducted and reported appropriately. The design of the experiments is simple, but nonetheless would have required much skill on the part of the researchers, especially to implement a design that provides an analogue to neurophysiological work in animal models. However, I have some concerns regarding the strength of certain conclusions and the choice of wording used in the paper. I suspect the authors can easily address my main concerns, which I outline below, and that, if they do, this manuscript would be a valuable contribution to the journal.

1. My biggest concern may be trivial: the most interesting data (exp 2) are shown time-locked to TMS-saccade asynchrony. The timing of the perceptual probes relative to TMS/saccade onset is not explicitly stated. Is it the same as in Experiment 1? If so, that means that the axes of Fig 3 can also be read as “target probe onset relative to saccade onset”. This information would be very useful. If the timing was different in Experiment 2, then the authors have a problem in which they could present their data relative to probe-saccade time OR TMS-saccade time, massively complicating the interpretability of the results.

Suggestion: the authors explicate the timing of perceptual probes in Exp 2. This should be done in the methods AND in the main text to improve clarity. I am hopeful that the authors implemented the more straightforward approach of time locking probes to TMS pulses, so the data are more straight forward to interpret, thereby rendering this concern trivial.

2. I do not see convincing evidence to support the conclusion that “presaccadic attention modulates perception through cortico-cortical feedback” (abstract, line 22). While the effects of TMS on FEF and V1/V2 show a causal role of those areas, a direction of connectivity has not been tested in this study. The hypothesis established by the authors, but not born out in the data (as far as I can see), is that, if pre-saccadic modulation results from FEF feedback to V1, early TMS at FEF should impact pre-saccadic

performance, whereas late TMS at V1/V2 should impact pre-saccadic performance. I agree that the cortico-cortical hypothesis is sensible and substantiated by converging lines of evidence, particularly from Moore et al., but was not directly confirmed in the present study.

Suggestion: The authors modify their wording to make clear that their evidence is consistent with this hypothesis but cannot confirm it. The study is sufficiently important that this softened conclusion will not diminish impact.

3. The conclusion “V1/V2 TMS did not affect presaccadic attention – unlike covert exogenous attention” (line 194, but similar statements appear throughout), is not correct. The authors show in Exp 2a that V1/V2 TMS does indeed alter presaccadic attention. As currently written, the manuscript has contradictory statements in different places – line 194 states V1/V2 TMS did not alter presaccadic attention, whereas 3 paragraphs later the authors state that it does. I appreciate that the time course of such effects complicates the matter, but that’s important – the time course of saccades is more easily defined than for shifts of exogenous attention, and so differences in time course can’t be a simple means to rule out similarities.

Suggestion: The authors reconcile these conflicting conclusions. Importantly, the authors have to outline what it means that V1/V2 TMS does indeed influence presaccadic attention similarly to (valid) exogenous cues.

4. The labels “presaccadic benefits” and “presaccadic costs” should be changed in order to use more precise language. In this context, it seems to me that a “benefit” should solely refer to an improvement in sensitivity, while a “cost” is a decrement in sensitivity, as a result of some manipulation. However, the authors use these terms to refer to their manipulation: benefits and costs are labels for sensitivity to perceptual probes presented at the saccade target vs opposite to the saccade target, respectively. This results in a confusing mess of words and data – Fig 3a shows a reduction in sensitivity following TMS under the “benefits” title, while Fig 3c shows an improvement in sensitivity following TMS under the “costs” title.

Suggestion: the authors choose more appropriate – and precise – language to label their panels, and describe their conditions so that the labels don’t confound the directionality of the results.

As a subpoint, I am not satisfied with the language of “presaccadic attention”. Much like the authors, I think there is enough evidence that suggests we should be cautious about assuming that changes in perception prior to a saccade invoke the exact same mechanisms as those involved in shifts of endogenous/exogenous attention. The continual use of the term “presaccadic attention”, therefore, may only add to the confusion. I don’t have a good suggestion to fix this, because “presaccadic

attention” is much more concise and convenient than “changes in perceptual sensitivity time locked to saccade onset”. Perhaps this point is broader than this one submission.

More minor issues:

5. I think the authors could mention somewhere in the main text that the overall different levels of sensitivity between valid vs invalid trials shouldn't be over-interpreted, because differently oriented stimuli were used in those conditions. This information is very clear in the methods, which I only read after much head scratching about why sensitivity is sometimes so different (or similar!) across conditions. I do not think this is an issue at all, and so perhaps the information could go in a figure caption (e.g. Fig 1) so as to not cause too much distraction. Sorry if I missed this somewhere.

6. I think in text the authors report Cohen's d to indicate effect size, which is great, but the “ d ” can be confused with the signal detection “ d' ” (d -prime). If the authors are indeed reporting Cohen's d , I suggest writing the full thing out to save confusion between the two “ d ” metrics.

7. I think the authors could comment on the apparent discrepancy in time course of sensitivity across TMS sites in the invalid, not-stimulated conditions. In the V1/V2 TMS condition, pre-saccadic sensitivity improves at the non-saccade location, which is surprising in and of itself, while it decreases in the FEF condition, which is in line with expectations. These conditions are the same, so it's not clear why there is such a large discrepancy. I cannot think of a simple explanation – anything that involves some interaction between the intermingled trial types would be highly speculative... but I think speculation is reasonable here, unless the authors have a more simple explanation.

8. Supp Figure 2 shows changing saccadic latencies as a function of TMS time. The motivation of this analysis is to understand whether TMS is influencing the saccade dynamics, rather than perceptual sensitivity per se. Another way of asking the same question is whether or not the saccadic latencies for different TMS timings come from one distribution, or multiple distributions. The results look like they support the latter – saccade onsets are slower when TMS occurred later. However, this is a probably incorrect interpretation, and I think the figure and explanation does the authors no favours here. Consider that the TMS-cue onset difference can only be large (~200ms) when the saccadic latency is sufficiently long so as to not cause the trial to be excluded. [Aside: the authors don't explicitly state that they exclude trials in which the perceptual probe is presented after saccade onset – they should add this information, which I hope is correct.] It is trivial to show that the same pattern of data as those shown in Supp Fig 2 are found using a single distribution of saccadic latencies, as well as the trial exclusion rule that saccadic latencies have to be longer than the cue-TMS offset. In matlab:

```
tmsTime = rand(1000,1)*200; % uniform distribution of TMS times
```

```
sacRTs = randn(1000,1)*100+230; % normal distribution of saccades with M = 230, SD = 100
```

```
i = sacRTs > tmsTime; % index only trials where saccade happened after TMS (other trials are excluded)
[muRT, binnedTMStime] = grpstats(sacRTs(i), round(tmsTime(i)/50)*50,{'mean','gname'}); % analyse sac
RTs in TMS bins
binnedTMStime = str2double(binnedTMStime);
scatter(binnedTMStime,muRT)
```

Great work.

Reviewer #2 (Remarks to the Author):

This paper uses TMS to explore the causal role of V1/V2 and the right FEF in pre-saccadic attention, and compares their results to previous studies, by the same lab, which have explored the role of the same brain regions in cover endogenous and exogenous attention. The results show that rFEF stimulation reduced presaccadic costs during the interval of saccade preparation, whereas V1/V2 stimulation reduced presaccadic benefits only shortly before saccade onset.

The paper is very interesting for the literature that explores the neural basis of spatial orienting. It is concisely and clearly written, and the experimental design and its findings are well presented in the results section.

I have a few recommendations and suggestions for clarification and improvement.

In the V1/V2 TMS condition, the internal control condition seems appropriate, as TMS generates a phosphene (some participants perceived phosphenes in the right visual field and others in the left visual field; and this data were collapsed). In the design, the cue can indicate the location where phosphenes are generated or the opposite location. But in the FEF stimulation, the authors assume a contralateral effect, which is not always the case in TMS studies of attentional regions. Without a control condition, where other brain region is stimulated, how can the authors be sure of the laterality effects of the right FEF TMS? I.e. what if right FEF stimulation produced effects ipsilaterally and not only contralaterally?

“Human neuroimaging studies indicate that saccade planning and covert exogenous and endogenous attention differentially modulate brain activity 35,51–53”.

Please provide references directly comparing saccade planning and cover orienting for this sentence.

A priori sample size calculation is not reported: “We chose a sample size in the range of previous psychophysics-TMS studies investigating presaccadic and covert attention 29,58,88,89”. Can the authors report a posteriori sensitivity for their sample size?

I do not understand this statement: “Observers performed 7 (Exp. 2a) or 10 (Exp. 2a) experimental blocks of 160 trials each, split into 2 experimental sessions”. I am guessing it refers to “7 (Exp. 2a) or 10 (Exp. 2b)”. If this is the case, why is the number of blocks different in the two experiments?

Reviewer #3 (Remarks to the Author):

Hanning, Fernández, and Carrasco investigated the differential roles of the human frontal eye field (FEF) and early visual cortex (V1/V2) in presaccadic attention, the brain mechanisms improving visual sensitivity at a target location (at a cost for other locations) already before a saccadic eye movement is made to the target location. To this end, they assessed visual sensitivity before saccadic eye movements using psychophysical and eye-tracking methods while at the same time manipulating the functioning of FEF or early visual cortex using transcranial magnetic stimulation (TMS). The results indicated that FEF stimulation reduced the costs of presaccadic attention for non-target locations, whereas the stimulation of early visual cortex reduced the benefits of presaccadic attention for target locations shortly before saccade onset.

The study deals with the important and timely research question of what brain mechanisms underlie the attentional improvements and impairments of visual perception that occur as a result of saccadic eye movements. The study has been designed and conducted with great technical rigor and great care, and overall, the results seem to support the conclusions. However, there are some points and suggestions (described in detail below) that should be addressed before the manuscript is published.

1) In Experiment 1, TMS applied to early visual cortex did not significantly affect presaccadic attention (p. 4, lines 99-100). This finding is contrasted with the results of a previous study (Fernández & Carrasco, 2020; Ref. 29) that found an extinction of covert exogenous attention by TMS in a similar paradigm. The contrasting findings from the two studies are interpreted as a dissociation of presaccadic attention and covert exogenous attention (e.g., p. 7, lines 195-197). However, this interpretation rests on a qualitative and informal comparison of the findings of the two studies (i.e. the presence of effects in one study and the absence of effects in the other study is interpreted as an interaction even though this is not tested statistically, cf. Nieuwenhuis et al. 2011). Therefore, to substantiate the interpretation in terms of a

dissociation, a quantitative analysis seems to be needed that compares the TMS effects on presaccadic and covert exogenous attention in a statistical fashion. Ideally, this would be done within the same experiment and the same observers. However, it could also be possible to do this using statistical analyses across the two studies, whereby special emphasis should be devoted to securing the required statistical power.

Reference

Nieuwenhuis, S., Forstmann, B. U., & Wagenmakers, E. J. (2011). Erroneous analyses of interactions in neuroscience: a problem of significance. *Nature Neuroscience*, 14(9), 1105-1107.

2) This point is similar to Point 1. The effects of TMS applied to early visual cortex and the frontal eye field are investigated in separate experiments and are analyzed separately (p. 5, Experiment 2a; p. 5-6, Experiment 2b), but are interpreted as differential effects of occipital and frontal areas in presaccadic attention (e.g., p. 7, lines 223-225). These effects should be compared in a quantitative fashion as well by performing statistical analyses across the experiments.

3) Some of the central findings of the study consist in the absence of statistical effects, for example that the TMS site did not interact with presaccadic attention in Experiment 1 (see above, see p. 4, lines 99-100) or that there were main effects and two-way interactions of TMS site, TMS time, and presaccadic attention in Experiment 2a (p. 5, lines 151-153). Interpreting null effects that are based on p-values (e.g. from ANOVAs, t-tests, etc.) has been criticized, because these statistical techniques are designed to detect the presence of effects but cannot speak to their absence (e.g., Rouder et al., 2009). Therefore, I recommend to provide Bayesian counterparts to all the reported statistical analyses, which aim to solve this problem and which allow to interpret evidence in favor of the null hypothesis (e.g., Bayes Factors, e.g., Rouder et al., 2009).

Reference

Rouder, J. N., Speckman, P. L., Sun, D., Morey, R. D., & Iverson, G. (2009). Bayesian t tests for accepting and rejecting the null hypothesis. *Psychonomic Bulletin & Review*, 16, 225-237.

4) The authors suggest that there had been a general alerting effect of the TMS on saccade latencies (p. 5, lines 133-138) and report the means (?) of the saccade latency distributions in Figure S2. Alerting might alter not only the central tendency of the distribution but also its shape. Thus, I think it would be helpful to characterize the distributions in more detail, also because the saccade latency affects the time that was available for presaccadic attention to build up.

5) In Figure 2b, observers' individual presaccadic attention effects are more heterogeneous as compared with the plotted exogenous attention effects. Why was this the case?

6) For the reported psychometric functions, I suggest to not only report the d_{max} parameter but also the other parameters and provide information about the goodness-of-fit.

Reviewer #1 (Remarks to the Author)

This study investigates the feedback mechanism between oculomotor structures and visual cortices in humans during the presaccadic period, during which visual sensitivity is enhanced at the saccade target and reduced at non-target locations. Using transcranial magnetic stimulation (TMS) on frontal (rFEF+) or visual (V1/V2) areas during saccade preparation and measuring perceptual performance, the authors found differential effects of rFEF+ and V1/V2 stimulation on presaccadic benefits and costs. By comparing the results with existing datasets, they further claim a dissociation of areas involved in presaccadic and covert attention.

The study is important, very well written and the results are presented clearly. The analyses in general are conducted and reported appropriately. The design of the experiments is simple, but nonetheless would have required much skill on the part of the researchers, especially to implement a design that provides an analogue to neurophysiological work in animal models. However, I have some concerns regarding the strength of certain conclusions and the choice of wording used in the paper. I suspect the authors can easily address my main concerns, which I outline below, and that, if they do, this manuscript would be a valuable contribution to the journal.

>> Thank you for your positive and helpful comments; addressing the technical and theoretical points you raised has improved our manuscript.

1. My biggest concern may be trivial: the most interesting data (exp 2) are shown time-locked to TMS-saccade asynchrony. The timing of the perceptual probes relative to TMS/saccade onset is not explicitly stated. Is it the same as in Experiment 1? If so, that means that the axes of Fig 3 can also be read as “target probe onset relative to saccade onset”. This information would be very useful. If the timing was different in Experiment 2, then the authors have a problem in which they could present their data relative to probe-saccade time OR TMS-saccade time, massively complicating the interpretability of the results.

Suggestion: the authors explicate the timing of perceptual probes in Exp 2. This should be done in the methods AND in the main text to improve clarity. I am hopeful that the authors implemented the more straightforward approach of time locking probes to TMS pulses, so the data are more straightforward to interpret, thereby rendering this concern trivial.

>> The timing of the probes was the same as in Exp. 1 (100-200ms ms after saccade cue onset); the timing of TMS pulses was varied. We chose to keep the probe timing constant –at a time point when saccade preparation has been shown to be reliably modulate perception (e.g., Montagnini & Castet, 2007; Rolfs & Carrasco, 2012; Li, Barbot & Carrasco, 2016; Hanning, Deubel, Szinte, 2019; Hanning, Himmelberg, Carrasco, 2022; Kwak, Hanning, Carrasco, 2023)– as our question of interest was when TMS affects presaccadic attention. Had we presented the probes at different time points during saccade preparation, we would have measured the time course of presaccadic attention (performance at the saccade target gradually increasing until saccade onset; e.g.: Deubel 2008, Hanning, Aagten-Murphy, Deubel, 2018) rather than the effect of V1/V2 or FEF+ stimulation on (deployed) presaccadic attention. We admit that an ideal design would independently vary both probe- and TMS pulse times – which we had considered but would have been too (time-)costly. We state the probe timing now more clearly in methods (line 365) and Fig. 1 legend.

2. I do not see convincing evidence to support the conclusion that “presaccadic attention modulates perception through cortico-cortical feedback” (abstract, line 22). While the effects of TMS on FEF and V1/V2 show a causal role of those areas, a direction of connectivity has not been tested in this study. The hypothesis established by the authors, but not born out in the data (as far as I can see), is that, if pre-saccadic modulation results from FEF feedback to V1, early TMS at FEF should impact pre-saccadic performance, whereas late TMS at V1/V2 should impact pre-saccadic performance. I agree that the cortico-cortical hypothesis is sensible and substantiated by converging lines of evidence, particularly from Moore et al., but was not directly confirmed in the present study.

Suggestion: The authors modify their wording to make clear that their evidence is consistent with this hypothesis but cannot confirm it. The study is sufficiently important that this softened conclusion will not diminish impact.

>> We have adjusted the wording accordingly (line 19).

3. The conclusion “V1/V2 TMS did not affect presaccadic attention – unlike covert exogenous attention” (line 194, but similar statements appear throughout), is not correct. The authors show in Exp 2a that V1/V2 TMS does indeed alter presaccadic attention. As currently written, the manuscript has contradictory statements in different places – line 194 states V1/V2 TMS did not alter presaccadic attention, whereas 3 paragraphs later the authors state that it does. I appreciate that the time course of such effects complicates the matter, but that’s important – the time course of saccades is more easily defined than for shifts of exogenous attention, and so differences in time course can’t be a simple means to rule out similarities.

Suggestion: The authors reconcile these conflicting conclusions. Importantly, the authors have to outline what it means that V1/V2 TMS does indeed influence presaccadic attention similarly to (valid) exogenous cues.

>> We removed the inconsistent statements, and we point out the similarities and differences between exogenous and presaccadic attention and describe the time at which they exert their effect (V1/V2 effect time-locked to saccade onset) more clearly (line 230).

4. The labels “presaccadic benefits” and “presaccadic costs” should be changed in order to use more precise language. In this context, it seems to me that a “benefit” should solely refer to an improvement in sensitivity, while a “cost” is a decrement in sensitivity, as a result of some manipulation. However, the authors use these terms to refer to their manipulation: benefits and costs are labels for sensitivity to perceptual probes presented at the saccade target vs opposite to the saccade target, respectively. This results in a confusing mess of words and data – Fig 3a shows a reduction in sensitivity following TMS under the “benefits” title, while Fig 3c shows an improvement in sensitivity following TMS under the “costs” title.

Suggestion: the authors choose more appropriate – and precise – language to label their panels, and describe their conditions so that the labels don’t confound the directionality of the results.

>> In line with previous work, we refer to the typically observed enhanced / decreased sensitivity at the saccade target / at non-targets *due to saccade preparation* (which is the “manipulation”) as “presaccadic benefits” and “presaccadic costs”, respectively. We modified the manuscript (line 162) to clarify this further. The observed of TMS on those two phenomena was (1) a reduction in

presaccadic benefits (i.e., less enhanced presaccadic performance at the saccade target) due to V1/V2-TSM, and (2) a reduction in presaccadic costs (i.e., less reduced / enhanced presaccadic performance at the non-saccade target). To avoid confusion, we changed the labels in Figure 3a to “reduced presaccadic benefit” and in Figure 3c to “reduced presaccadic cost”.

As a subpoint, I am not satisfied with the language of “presaccadic attention”. Much like the authors, I think there is enough evidence that suggests we should be cautious about assuming that changes in perception prior to a saccade invoke the exact same mechanisms as those involved in shifts of endogenous/exogenous attention. The continual use of the term “presaccadic attention”, therefore, may only add to the confusion. I don’t have a good suggestion to fix this, because “presaccadic attention” is much more concise and convenient than “changes in perceptual sensitivity time locked to saccade onset”. Perhaps this point is broader than this one submission.

>> The term presaccadic attention is well established and does not imply the same mechanism to other types of attention; Covert endogenous and exogenous attention also rely on different mechanisms but are both still referred to as forms of attention; “Changes in perceptual sensitivity time locked to saccade onset” are typically referred to as (consequences of) presaccadic attention – which does not imply identical mechanisms as other types of attention. We have recently published a review on this topic (To look or not to look: Dissociating presaccadic and covert spatial attention, TINS, 2022, Li, Hanning & Carrasco, 2022). To further clarify, we have extended the definition of attention (**line 31**).

More minor issues:

5. I think the authors could mention somewhere in the main text that the overall different levels of sensitivity between valid vs invalid trials shouldn’t be over-interpreted, because differently oriented stimuli were used in those conditions. This information is very clear in the methods, which I only read after much head scratching about why sensitivity is sometimes so different (or similar!) across conditions. I do not think this is an issue at all, and so perhaps the information could go in a figure caption (e.g. Fig 1) so as to not cause too much distraction. Sorry if I missed this somewhere.

>> We have added this information to the Figure 1 legend.

6. I think in text the authors report Cohen’s d to indicate effect size, which is great, but the “ d ” can be confused with the signal detection “ d ” (d -prime). If the authors are indeed reporting Cohen’s d , I suggest writing the full thing out to save confusion between the two “ d ” metrics.

>> We now spell out “Cohen’s d ” throughout the manuscript.

7. I think the authors could comment on the apparent discrepancy in time course of sensitivity across TMS sites in the invalid, not-stimulated conditions. In the V1/V2 TMS condition, pre-saccadic sensitivity improves at the non-saccade location, which is surprising in and of itself, while it decreases in the FEF condition, which is in line with expectations. These conditions are the same, so it’s not clear why there is such a large discrepancy. I cannot think of a simple explanation – anything that involves some interaction between the intermingled trial types would be highly speculative... but I think speculation is reasonable here, unless the authors have a more simple explanation.

>> We have evaluated the discrepancy pointed out in the sensitivity time course across TMS sites in the invalid, not-stimulated conditions; a 2 (experiment / stimulation site: V1/V2 vs. rFEF+) x 4 (stimulation time) mixed effects ANOVA showed no interaction between time and stimulation site ($F(3,42)=1.76, p=0.170$), merely the main effect of experiment (stimulation site) was significant ($F(3,42)=5.37, p=0.036$).

This overall sensitivity difference in the invalid conditions of the two experiments / stimulation sites can be explained by our tilt angle titration / adjustment procedure, which we now explain more in detail (**line 390**). If during Experiment 2 the overall discrimination performance in valid or invalid trials (across hemifields) deviated more than $\pm 10\%$ from 80% ($d' \sim 2$), we slightly adjusted the tilt angle of the respective condition accordingly. Given that in Experiment 2b rFEF+ stimulation increased performance at the stimulated side (and thus overall performance), the resulting tilt angle adjustment (to keep overall task difficulty in the invalid condition around 80%) could have lowered performance at the unstimulated side.

To speculate about the diverging (but non-significant) trend in time course of sensitivity across TMS sites in the invalid, not-stimulated conditions: We might see a push-pull mechanism here; e.g. in Exp.2a (V1/V2 stimulation) sensitivity at the saccade targets in the stimulated hemifield is reduced for stimulation right before saccade onset, leaving more (attentional) resources for other locations, such as the (invalid) location opposite the saccade target (in the non-stimulated hemifield). In Exp.2b rFEF+ stimulation did not affect sensitivity at the saccade target, causing no such “enhancement” at the invalid location (in the non-stimulated hemifield).

8. Supp Figure 2 shows changing saccadic latencies as a function of TMS time. The motivation of this analysis is to understand whether TMS is influencing the saccade dynamics, rather than perceptual sensitivity per se. Another way of asking the same question is whether or not the saccadic latencies for different TMS timings come from one distribution, or multiple distributions. The results look like they support the latter – saccade onsets are slower when TMS occurred later. However, this is a probably incorrect interpretation, and I think the figure and explanation does the authors no favours here. Consider that the TMS-cue onset difference can only be large (~ 200 ms) when the saccadic latency is sufficiently long so as to not cause the trial to be excluded. [Aside: the authors don't explicitly state that they exclude trials in which the perceptual probe is presented after saccade onset – they should add this information, which I hope is correct.] It is trivial to show that the same pattern of data as those shown in Supp Fig 2 are found using a single distribution of saccadic latencies, as well as the trial exclusion rule that saccadic latencies have to be longer than the cue-TMS offset. In matlab:

```
tmsTime = rand(1000,1)*200; % uniform distribution of TMS times
sacRTs = randn(1000,1)*100+230; % normal distribution of saccades with M = 230, SD = 100
i = sacRTs > tmsTime; % index only trials where saccade happened after TMS (other trials are excluded)
[muRT, binnedTMStime] = grpstats(sacRTs(i), round(tmsTime(i)/50)*50,{'mean','gname'}); % analyse sac RTs in TMS bins
binnedTMStime = str2double(binnedTMStime);
scatter(binnedTMStime,muRT)
```

>> First, we want to confirm that trials in which the perceptual probe was presented after saccade onset were of course excluded from further analysis. We had provided this information in Methods – Eye data preprocessing, final sentence (**line 402**).

As we have clarified in response to comment 1, the probe timing was constant (100ms after cue onset), whereas TMS pulse timing was varied (0-200ms after cue onset). Given that probe and TMS-pulse were not coupled, trials with large TMS-cue onset difference were not more likely to be excluded and the presented results are not a consequence of trial exclusion.

Great work.

>> Thanks again for your thoughtful and helpful comments

Reviewer #2 (Remarks to the Author)

This paper uses TMS to explore the causal role of V1/V2 and the right FEF in pre-saccadic attention, and compares their results to previous studies, by the same lab, which have explored the role of the same brain regions in cover endogenous and exogenous attention. The results show that rFEF stimulation reduced presaccadic costs during the interval of saccade preparation, whereas V1/V2 stimulation reduced presaccadic benefits only shortly before saccade onset.

The paper is very interesting for the literature that explores the neural basis of spatial orienting. It is concisely and clearly written, and the experimental design and its findings are well presented in the results section.

I have a few recommendations and suggestions for clarification and improvement.

>> Thank you for your helpful comments; addressing them has improved our manuscript.

1) In the V1/V2 TMS condition, the internal control condition seems appropriate, as TMS generates a phosphene (some participants perceived phosphenes in the right visual field and others in the left visual field; and this data were collapsed). In the design, the cue can indicate the location where phosphenes are generated or the opposite location. But in the FEF stimulation, the authors assume a contralateral effect, which is not always the case in TMS studies of attentional regions. Without a control condition, where other brain region is stimulated, how can the authors be sure of the laterality effects of the right FEF TMS? I.e. what if right FEF stimulation produced effects ipsilaterally and not only contralaterally?

>> We assume a hemifield-specific, contralateral effect of rFEF+ stimulation as we have previously shown (Fernández, Hanning, & Carrasco, 2023, PNAS) that with the same neuro-navigated rFEF+ localization and stimulation protocol (timing and intensity) TMS only affected perceptual modulations of covert endogenous in the contralateral hemifield, leaving the known effects of endogenous attention on visual performance in the ipsilateral hemifield unaffected. Additionally, several studies focused on the role of FEF+ on attentional modulation have found that only the right FEF+ affects behavior (Hung, Driver & Walsh, 2011; Ronconi et al., 2014; Esterman et al., 2015). Some others have found diminished lateralized effects (Chanes, et al., 2012). Therefore, it is possible that previous lateralized effects are dependent on task and stimulus parameters.

2) “Human neuroimaging studies indicate that saccade planning and covert exogenous and endogenous attention differentially modulate brain activity 35,51–53”.

Please provide references directly comparing saccade planning and cover orienting for this sentence.

>> We now provide references directly comparing covert exogenous / endogenous attention and saccade planning (**line 60**):

Beauchamp, M. S., Petit, L., Ellmore, T. M., Ingeholm, J., & Haxby, J. V. (2001). A parametric fMRI study of overt and covert shifts of visuospatial attention. *Neuroimage*, 14(2), 310-321.

Chica, A.B., Bartolomeo, P., and Lupiáñez, J. (2013). Two cognitive and neural systems for endogenous and exogenous spatial attention. *Behavioural Brain Research* 237, 107–123. 10.1016/j.bbr.2012.09.027.

- Dugué, L., Merriam, E.P., Heeger, D.J., and Carrasco, M. (2020). Differential impact of endogenous and exogenous attention on activity in human visual cortex. *Sci Rep-uk* 10, 21274. 10.1038/s41598-020-78172-x.
- Fairhall, S. L., Indovina, I., Driver, J., & Macaluso, E. (2009). The brain network underlying serial visual search: comparing overt and covert spatial orienting, for activations and for effective connectivity. *Cerebral cortex*, 19(12), 2946-2958.
- Konen, C. S., Kleiser, R., Bremmer, F., & Seitz, R. J. (2007). Different cortical activations during visuospatial attention and the intention to perform a saccade. *Experimental brain research*, 182, 333-341.
- Perry, R. J., & Zeki, S. (2000). The neurology of saccades and covert shifts in spatial attention: an event-related fMRI study. *Brain*, 123(11), 2273-2288.
- Wu, T., Mackie, M. A., Chen, C., & Fan, J. (2022). Representational coding of overt and covert orienting of visuospatial attention in the frontoparietal network. *NeuroImage*, 261, 119499.

3) A priori sample size calculation is not reported: “We chose a sample size in the range of previous psychophysics-TMS studies investigating presaccadic and covert attention 29,58,88,89”. Can the authors report a posteriori sensitivity for their sample size?

>>We now provide an estimate of statistical power via Bayes Factor and Bayesian Information Criterion probabilities throughout the manuscript.

4) I do not understand this statement: “Observers performed 7 (Exp. 2a) or 10 (Exp. 2a) experimental blocks of 160 trials each, split into 2 experimental sessions”. I am guessing it refers to “7 (Exp. 2a) or 10 (Exp. 2b)”. If this is the case, why is the number of blocks different in the two experiments?

>> Yes, this indeed should read “7 (Exp. 2a) or 10 (Exp. 2b)”, we have corrected this typo (**line 370**). We also added an explanation for the different number of blocks (**line 371**): Because stimulating oculomotor area rFEF+ (in Exp 2b) might have affected saccade parameters (which, with our stimulation protocol turned out not to be the case), we had planned to collect a higher number of trials (as compared to Exp. 2a with V1/V2 stimulation, for which we knew from Exp. 1 that saccade parameters are not affected).

[Reviewer #3 (Remarks to the Author)]

Hanning, Fernández, and Carrasco investigated the differential roles of the human frontal eye field (FEF) and early visual cortex (V1/V2) in presaccadic attention, the brain mechanisms improving visual sensitivity at a target location (at a cost for other locations) already before a saccadic eye movement is made to the target location. To this end, they assessed visual sensitivity before saccadic eye movements using psychophysical and eye-tracking methods while at the same time manipulating the functioning of FEF or early visual cortex using transcranial magnetic stimulation (TMS). The results indicated that FEF stimulation reduced the costs of presaccadic attention for non-target locations, whereas the stimulation of early visual cortex reduced the benefits of presaccadic attention for target locations shortly before saccade onset.

The study deals with the important and timely research question of what brain mechanisms underlie the attentional improvements and impairments of visual perception that occur as a result of saccadic eye movements. The study has been designed and conducted with great technical rigor and great care, and overall, the results seem to support the conclusions. However, there are some points and suggestions (described in detail below) that should be addressed before the manuscript is published.

>> Thank you for your helpful feedback, the suggested additional analyses have strengthened our manuscript.

1) In Experiment 1, TMS applied to early visual cortex did not significantly affect presaccadic attention (p. 4, lines 99-100). This finding is contrasted with the results of a previous study (Fernández & Carrasco, 2020; Ref. 29) that found an extinction of covert exogenous attention by TMS in a similar paradigm. The contrasting findings from the two studies are interpreted as a dissociation of presaccadic attention and covert exogenous attention (e.g., p. 7, lines 195-197). However, this interpretation rests on a qualitative and informal comparison of the findings of the two studies (i.e. the presence of effects in one study and the absence of effects in the other study is interpreted as an interaction even though this is not tested statistically, cf. Nieuwenhuis et al. 2011). Therefore, to substantiate the interpretation in terms of a dissociation, a quantitative analysis seems to be needed that compares the TMS effects on presaccadic and covert exogenous attention in a statistical fashion. Ideally, this would be done within the same experiment and the same observers. However, it could also be possible to do this using statistical analyses across the two studies, whereby special emphasis should be devoted to securing the required statistical power.

Reference

Nieuwenhuis, S., Forstmann, B. U., & Wagenmakers, E. J. (2011). Erroneous analyses of interactions in neuroscience: a problem of significance. *Nature Neuroscience*, 14(9), 1105-1107.

>> Agreed. We verified the dissociation in the existing data using a 2 (exogenous vs. presaccadic attention, between subject factor) * 3 (valid, neutral, invalid) * 2 (test stimulated/not stimulated) mixed ANOVA, which showed a 3-way interaction. We added this result to the main text (**line 118**).

2) This point is similar to Point 1. The effects of TMS applied to early visual cortex and the frontal eye field are investigated in separate experiments and are analyzed separately (p. 5, Experiment 2a; p. 5-6, Experiment 2b), but are interpreted as differential effects of occipital and frontal areas in presaccadic attention (e.g., p. 7, lines 223-225). These effects should be compared in a quantitative fashion as well by performing statistical analyses across the experiments.

>> We also confirmed this dissociation by conducting a 2 (stimulation site: V1/V2 vs. rFEF+ stimulation) * 2 (attention condition: valid, invalid) * 4 (stimulation time) repeated measures ANOVA, in which we compared presaccadic benefits and costs at the V1/V2 and rFEF+ stimulated hemifield. We conducted this comparison across experiments for the 6 observers for whom we collected data for both stimulation sites (line 207).

3) Some of the central findings of the study consist in the absence of statistical effects, for example that the TMS site did not interact with presaccadic attention in Experiment 1 (see above, see p. 4, lines 99-100) or that there were main effects and two-way interactions of TMS site, TMS time, and presaccadic attention in Experiment 2a (p. 5, lines 151-153). Interpreting null effects that are based on p-values (e.g. from ANOVAs, t-tests, etc.) has been criticized, because these statistical techniques are designed to detect the presence of effects but cannot speak to their absence (e.g., Rouder et al., 2009). Therefore, I recommend to provide Bayesian counterparts to all the reported statistical analyses, which aim to solve this problem and which allow to interpret evidence in favor of the null hypothesis (e.g., Bayes Factors, e.g., Rouder et al., 2009).

Reference

Rouder, J. N., Speckman, P. L., Sun, D., Morey, R. D., & Iverson, G. (2009). Bayesian t tests for accepting and rejecting the null hypothesis. *Psychonomic Bulletin & Review*, 16, 225-237.

>> We now follow-up null effects with Bayesian statistics throughout the manuscript.

4) The authors suggest that there had been a general alerting effect of the TMS on saccade latencies (p. 5, lines 133-138) and report the means (?) of the saccade latency distributions in Figure S2. Alerting might alter not only the central tendency of the distribution but also its shape. Thus, I think it would be helpful to characterize the distributions in more detail, also because the saccade latency affects the time that was available for presaccadic attention to build up.

>> We report the group-average (mean) of individual participants' median saccade latency (same for saccade precision, we added this information to the legend of Fig. S2).

We attached to this review (final page) latency histograms for each Experiment (V1/V2 stimulation & rFEF+ stimulation), saccade direction (to stimulated side, to ipsilateral side), as well as stimulation time bin (0-200ms, as in Figure S2); both split for observers and combined (last column).

Because there is no visible difference between the two saccade directions (to / away from the stimulated hemifield) – which would indicate a TMS specific effect (that could explain / contribute to our observed effects on presaccadic attention) – for any time bin, neither at the group level, nor a consistent pattern for individual observers, we prefer not to include these plots in the manuscript.

5) In Figure 2b, observers' individual presaccadic attention effects are more heterogeneous as compared with the plotted exogenous attention effects. Why was this the case?

>> We attribute the bigger between-participant variance to overall more pronounced effect of presaccadic attention ($d' \sim 2.5$) as compared to covert exogenous and endogenous attention ($d' \sim 1$). Moreover, the effect of presaccadic attention is known to rise until saccade onset. In our experiment we do not explore this time course but present the test stimuli at a fixed time (100-200ms after saccade cue onset). Inter-individual differences in saccade latency cause the test stimulus to be presented closer to or earlier before saccade onset – in which case a different magnitude of presaccadic benefits and costs are expected.

6) For the reported psychometric functions, I suggest to not only report the d_{max} parameter but also the other parameters and provide information about the goodness-of-fit.

>> We now also report C_{50} and the goodness-of-fit (**lines 101 & 417**).

Saccade latency histograms
Experiment 2a: V1/V2 stimulation

saccade to stimulated side
saccade to non-stimulated side

group data

Saccade latency histograms
Experiment 2b: rFEF+ stimulation

saccade to stimulated side
saccade to non-stimulated side

group data

REVIEWERS' COMMENTS

Reviewer #1 (Remarks to the Author):

I commend the authors on their thorough responses to my earlier review. They have done a great job at clarifying several of my misunderstandings, and the changes in the manuscript will hopefully mitigate the potential for others to have similar misunderstandings.

I have only one remaining minor comment. In my previous comment (7), I asked the authors to speculate on performance differences in the two unstimulated conditions at the non-saccade target locations. The authors response is convincing (or at the very least, plausible), but as far as I can tell they did not make any changes to the manuscript to explain to a naive reader why these stark differences in performance may have come about. I encourage the authors to copy some of their helpful response to me into their results or discussion.

Reviewer #2 (Remarks to the Author):

The authors have adequately responded to all my comments although I do not agree 100% with their response to this comment.

“- In the V1/V2 TMS condition, the internal control condition seems appropriate, as TMS generates a phosphene (some participants perceived phosphenes in the right visual field and others in the left visual field; and this data were collapsed). In the design, the cue can indicate the location where phosphenes are generated or the opposite location. But in the FEF stimulation, the authors assume a contralateral effect, which is not always the case in TMS studies of attentional regions. Without a control condition, where other brain region is stimulated, how can the authors be sure of the laterality effects of the right FEF TMS? I.e. what if right FEF stimulation produced effects ipsilaterally and not only contralaterally?

>> We assume a hemifield-specific, contralateral effect of rFEF+ stimulation as we have previously shown (Fernández, Hanning, & Carrasco, 2023, PNAS) that with the same neuro-navigated rFEF+ localization and stimulation protocol (timing and intensity) TMS only affected perceptual modulations of covert endogenous in the contralateral hemifield, leaving the known effects of endogenous attention on visual performance in the ipsilateral hemifield unaffected.

Additionally, several studies focused on the role of FEF+ on attentional modulation have found that only the right FEF+ affects behavior (Hung, Driver & Walsh, 2011; Ronconi et al., 2014; Esterman et al., 2015).

Some others have found diminished lateralized effects (Chanes, et al., 2012). Therefore, it is possible that previous lateralized effects are dependent on task and stimulus parameters”.

Although I do agree in the fact that in general TMS-right FEF effects are contralateral, it is not always the case (see e.g. Silvanto et al 2006, J. Neurophysiol: “TMS of the right FEF changed the sensitivity of left and right MT/V5 whereas TMS of the left FEF changed the sensitivity only of the left MT/V5”). I think it will be fair to state in the manuscript that this paradigm assumes contralateral effects, and accept the limitation of the results if TMS effects were bilateral.

Reviewer #3 (Remarks to the Author):

The authors addressed all of the reviewers' comments thoroughly and successfully, so that I don't have any further suggestions.

Reviewer #1 (Remarks to the Author):

I commend the authors on their thorough responses to my earlier review. They have done a great job at clarifying several of my misunderstandings, and the changes in the manuscript will hopefully mitigate the potential for others to have similar misunderstandings.

I have only one remaining minor comment. In my previous comment (7), I asked the authors to speculate on performance differences in the two unstimulated conditions at the non-saccade target locations. The authors response is convincing (or at the very least, plausible), but as far as I can tell they did not make any changes to the manuscript to explain to a naive reader why these stark differences in performance may have come about. I encourage the authors to copy some of their helpful response to me into their results or discussion.

Thank you again for your helpful feedback, addressing your comments has strengthened our manuscript and we have now also included an explanation for the observed performance differences in the two unstimulated conditions at the non-saccade target (**lines 209–221**).

Reviewer #2 (Remarks to the Author):

The authors have adequately responded to all my comments although I do not agree 100% with their response to this comment.

“- In the V1/V2 TMS condition, the internal control condition seems appropriate, as TMS generates a phosphene (some participants perceived phosphenes in the right visual field and others in the left

visual field; and this data were collapsed). In the design, the cue can indicate the location where phosphenes are generated or the opposite location. But in the FEF stimulation, the authors assume a contralateral effect, which is not always the case in TMS studies of attentional regions. Without a control condition, where other brain region is stimulated, how can the authors be sure of the laterality effects of the right FEF TMS? I.e. what if right FEF stimulation produced effects ipsilaterally and not only contralaterally?

>> We assume a hemifield-specific, contralateral effect of rFEF+ stimulation as we have previously shown (Fernández, Hanning, & Carrasco, 2023, PNAS) that with the same neuro-navigated rFEF+ localization and stimulation protocol (timing and intensity) TMS only affected perceptual modulations of covert endogenous in the contralateral hemifield, leaving the known effects of endogenous attention on visual performance in the ipsilateral hemifield unaffected. Additionally, several studies focused on the role of FEF+ on attentional modulation have found that only the right FEF+ affects behavior (Hung, Driver & Walsh, 2011; Ronconi et al., 2014; Esterman et al., 2015). Some others have found diminished lateralized effects (Chanes, et al., 2012). Therefore, it is possible that previous lateralized effects are dependent on task and stimulus parameters”.

Although I do agree in the fact that in general TMS-right FEF effects are contralateral, it is not always the case (see e.g. Silvanto et al 2006, J. Neurophysiol: “TMS of the right FEF changed the sensitivity of left and right MT/V5 whereas TMS of the left FEF changed the sensitivity only of the left MT/V5”). I think it will be fair to state in the manuscript that this paradigm assumes contralateraleffects, and accept the limitation of the results if TMS effects were bilateral.

We now state in the manuscript why our protocol assumes contralateral effects and in this context also cite Silvanto et al. 2006 (line 142–148). Thank you again for your feedback.

Reviewer #3 (Remarks to the Author):

The authors addressed all of the reviewers' comments thoroughly and successfully, so that I don't have any further suggestions.

Thank you again for the helpful feedback!